# Interpretation of the Epigenetic Signature of Facioscapulohumeral Muscular Dystrophy in Light of Genotype-Phenotype Studies

**DOI:** 10.3390/ijms21072635

**Published:** 2020-04-10

**Authors:** Ana Nikolic, Takako I Jones, Monica Govi, Fabiano Mele, Louise Maranda, Francesco Sera, Giulia Ricci, Lucia Ruggiero, Liliana Vercelli, Simona Portaro, Luisa Villa, Chiara Fiorillo, Lorenzo Maggi, Lucio Santoro, Giovanni Antonini, Massimiliano Filosto, Maurizio Moggio, Corrado Angelini, Elena Pegoraro, Angela Berardinelli, Maria Antonetta Maioli, Grazia D’Angelo, Antonino Di Muzio, Gabriele Siciliano, Giuliano Tomelleri, Maurizio D’Esposito, Floriana Della Ragione, Arianna Brancaccio, Rachele Piras, Carmelo Rodolico, Tiziana Mongini, Frederique Magdinier, Valentina Salsi, Peter L. Jones, Rossella Tupler

**Affiliations:** 1Department of Science of Life, Institute of Biology, University of Modena and Reggio Emilia, 41125 Modena, Italy; ananikolic_bg@yahoo.com (A.N.); monicagovi@hotmail.it (M.G.); valentina.salsi@unimore.it (V.S.); 2Department of Pharmacology, School of Medicine, University of Nevada, Reno, NV 89557, USA; takakojones@med.unr.edu (T.I.J.); peterjones@med.unr.edu (P.L.J.); 3Center for Genome Research, University of Modena and Reggio Emilia, 41125 Modena, Italy; fabiano.mele@unimore.it; 4Department of Quantitative Health Sciences, University of Massachusetts Medical School, Worcester, MA 01605, USA; Louise.Maranda@umassmed.edu; 5Department of Public Health, Environments and Society, London School of Hygiene and Tropical Medicine, London WC1E 7HT, UK; francesco.sera@lshtm.ac.uk; 6Department of Clinical and Experimental Medicine, Neurological Clinic, 56126 Pisa, Italy; giulia.ricci@med.unipi.it (G.R.); gabriele.siciliano@unipi.it (G.S.); 7Department of Neurosciences and Reproductive and Odontostomatologic Sciences, University Federico II, 80137 Naples, Italy; lucia.ruggiero@unina.it (L.R.); lucio.santoro@unina.it (L.S.); 8Department of Neurosciences “Rita Levi Montalcini”, University of Turin, 10124 Turin, Italy; liliana.vercelli@unito.it (L.V.); tizianaenrica.mongini@unito.it (T.M.); 9Department of Neuroscience, Mental Health and Sensory Organs, S. Andrea Hospital, University of Rome “Sapienza”, 00185 Rome, Italy; simonaportaro@hotmail.it (S.P.); giovanni.antonini@uniroma1.it (G.A.); 10Department of Neuroscience, Foundation IRCCS Ca’ Granda, Ospedale Maggiore Policlinico, 20122 Milan, Italy; villaluisa@hotmail.com (L.V.); maurizio.moggio@unimi.it (M.M.); 11Pediatric Neurology and Neuromuscular Disorders Unit, Department of Neurosciences, Rehabilitation, Ophthalmology, Genetics, Maternal and Child Health, University of Genoa, 16126 Genoa, Italy; chiara.fiorillo@edu.unige.it; 12IRCCS Foundation, C. Besta Neurological Institute, 20133 Milan, Italy; lorenzo.maggi@istituto-besta.it; 13Neurology Clinic, ‘‘Spedali Civili’’. Hospital, 25123 Brescia, Italy; massimiliano.filosto@unibs.it; 14Ospedale S.Camillo IRCCS, Lido di Venezia, 20126 Venezia, Italy; corrado.angelini@ospedalesancamillo.net; 15Department of Neurosciences, University of Padua, 35128 Padua, Italy; elena.pegoraro@unipd.it; 16Neurology and Psychiatry, IRCCS Institute ‘C.Mondino’ Foundation, 27100 Pavia, Italy; angela.berardinelli@mondino.it; 17ASL8, Centro Sclerosi Multipla, 09126 Cagliari, Italy; ma.maioli@tiscali.it (M.A.M.); rachele_piras@tiscali.it (R.P.); 18Department of Neurorehabilitation, IRCCS Institute Eugenio Medea, 23842 Bosisio Parini, Italy; grazia.dangelo@bp.Inf.it; 19Center for Neuromuscular Disease, CeSI, University ‘‘G. D’Annunzio’’, 66100 Chieti, Italy; antonino.dimuzio1@tin.it; 20Department of Biomedical, Metabolic and Neural Sciences, University of Modena and Reggio Emilia, 41125 Modena, Italy; giuliano.tomelleri@univr.it; 21Institute of Genetics and Biophysics, A. Buzzati Traverso, IGB, Consiglio Nazionale delle Ricerche, 80131 Naples, Italy; maurizio.desposito@igb.cnr.it (M.D.); floriana.dellaragione@igb.cnr.it (F.D.R.); a.brancaccio@hsantalucia.it (A.B.); 22Department of Clinical and Experimental Medicine, University of Messina, 98124 Messina, Italy; crodolico@unime.it; 23Aix Marseille Univ, INSERM, MMG, U 1251, 13005 Marseille, France; Frederique.MAGDINIER@univ-amu.fr; 24Center for Neuroscience and Neurotechnology, University of Modena and Reggio Emilia, 41125 Modena, Italy; 25Department of Molecular Cell and Cancer Biology, University of Massachusetts Medical School, Worcester, MA 01605, USA; 26Li Weibo Institute for Rare Diseases Research at the University of Massachusetts Medical School, Worcester, MA 01605, USA

**Keywords:** FSHD, D4Z4 reduced allele, DNA methylation, genotype–phenotype correlation, molecular diagnosis

## Abstract

Facioscapulohumeral muscular dystrophy (FSHD) is characterized by incomplete penetrance and intra-familial clinical variability. The disease has been associated with the genetic and epigenetic features of the D4Z4 repetitive elements at 4q35. Recently, D4Z4 hypomethylation has been proposed as a reliable marker in the FSHD diagnosis. We exploited the Italian Registry for FSHD, in which FSHD families are classified using the Clinical Comprehensive Evaluation Form (CCEF). A total of 122 index cases showing a classical FSHD phenotype (CCEF, category A) and 110 relatives were selected to test with the receiver operating characteristic (ROC) curve, the diagnostic and predictive value of D4Z4 methylation. Moreover, we performed DNA methylation analysis in selected large families with reduced penetrance characterized by the co-presence of subjects carriers of one D4Z4 reduced allele with no signs of disease or presenting the classic FSHD clinical phenotype. We observed a wide variability in the D4Z4 methylation levels among index cases revealing no association with clinical manifestation or disease severity. By extending the analysis to family members, we revealed the low predictive value of D4Z4 methylation in detecting the affected condition. In view of the variability in D4Z4 methylation profiles observed in our large cohort, we conclude that D4Z4 methylation does not mirror the clinical expression of FSHD. We recommend that measurement of this epigenetic mark must be interpreted with caution in clinical practice.

## 1. Introduction

Facioscapulohumeral muscular dystrophy (FSHD) (OMIM#158900) is characterized by insidious onset and progressive wasting of a highly selective set of muscle groups [1]. FSHD is the third most common hereditary myopathy with a prevalence of 1 in 20,000 [2]. It has been historically classified as a fully penetrant autosomal dominant disease [1], even though recent genotype–phenotype studies revealed reduced penetrance of disease, which can vary substantially among different families [3,4,5].

FSHD has been genetically linked to the reduction of an integral number of tandem 3.3-kilobase D4Z4 repeat units (RU) located on chromosome 4q35 [6,7]. Although nearly identical D4Z4 sequences reside on chromosome 10q26 [8], only subjects with a D4Z4 reduced array (DRA) on chromosome 4, but not on chromosome 10, develop FSHD [9]. The size of the D4Z4 arrays on chromosomes 4 and 10 is assessed by digestion with *EcoR*I, which cuts outside the repeat array, followed by pulse-field gel electrophoresis (PFGE) and Southern blotting [10]. Based on these results, *EcoR*I alleles larger than 50 kb (≥11 D4Z4 RU) originating from chromosome 4q have been considered normal, whereas alleles of 35 kb or shorter (≤8 D4Z4 RU) have been considered diagnostic for FSHD and considered highly sensitive and specific [11,12]. Through the years the threshold size of D4Z4 alleles has been increased from the original 28 kb (6 RU) [6] to 35 kb (8 RU) [7], with FSHD cases carrying D4Z4 alleles of 38-41 kb (9–11 RU) considered borderline alleles [13]. To complicate FSHD molecular diagnosis, several genotype–phenotype studies have shown that D4Z4 alleles with 4-8 RU are also present in 3% of the general population [4,10,14]. Moreover, 5–10% of FSHD patients carry D4Z4 arrays of size within the range of the general healthy population (11 RU or more) on both 4q chromosomes [14,15]. Molecularly, these subjects represent a second form of the disease, FSHD2. Therefore, the number of D4Z4 RU at 4q35 does not *per se* characterize the disease. As a matter of fact, genotype–phenotype studies have shown a large spectrum of clinical phenotypes in myopathic subjects carrying a D4Z4 reduced allele as well as reduced penetrance among relatives carrying the same D4Z4 reduced allele [5,13,16,17,18,19]. This great heterogeneity has generated the need for additional markers to support FSHD diagnosis, genetic counseling, and patients’ stratification for clinical trials. 

From this perspective, it has been shown that reduction of DNA methylation of the 4q35 D4Z4 array characterizes both FSHD1 and FSHD2 [20,21]. Furthermore, roughly 85% of FSHD2 patients carry heterozygous dominant mutations in the *SMCHD1* (*structural maintenance of chromosomes flexible hinge domain containing-1*) gene [22], which encodes an epigenetic regulator of specific loci including D4Z4 arrays. It has been thus proposed that repeat contraction (FSHD1) or *SMCHD1* mutation (FSHD2) is involved in chromatin changes resulting in reduced D4Z4 DNA methylation, which characterizes disease. It has thus been suggested that molecular diagnosis of FSHD can be based on genetic (presence or absence of DRA) and epigenetic (D4Z4 hypomethylation and/or *SMCHD1* variant) characteristics. 

The methylation status of the D4Z4 arrays can be measured by using methylation-sensitive restriction enzyme (MSRE) analysis, which measures the methylation status of a single CpG dinucleotide in the proximal-most D4Z4 RU of the 4q and 10q arrays [22,23,24], or using sodium bisulfite sequencing (BSS), which measures the percentage of methylated CpG dinucleotides across a region either within each D4Z4 RU [25,26,27] or specifically in the distal-most 4q35 D4Z4 RU [27].

Here, we present the results of a large consecutive study of D4Z4 methylation on index cases presenting classical features of FSHD and their relatives to test the association of genetic, epigenetic, and clinical features. All subjects were clinically examined by applying the Comprehensive Clinical Evaluation Form (CCEF), which assesses the degree of motor impairment, describes the observed phenotypic features, and captures the wide clinical spectrum of FSHD families [19]. D4Z4 Methylation status has been measured using MSRE analysis in all subjects and BSS of the distal-most 4q35 RU in selected families.

## 2. Results

### 2.1. A Highly Variable Range of D4Z4 Methylation Level in FSHD ndex Cases

We investigated D4Z4 methylation status at 4q35 and 10q26 through MSRE1 in 122 FSHD1 index cases with FSHD clinical score ≥1 at the time of evaluation (age at examination ≥10 years). As shown in Figure 1A and Appendix A, we observed a highly variable distribution of D4Z4 methylation level ranging from 3% to 74%. Remarkably, through MRSE2 assay, Figure 1B, we found that 87 out of 122 FSHD index cases (71%) displayed a level of methylation higher than 25% on chromosome 4q35. Statistical analysis showed no significant association between the MSRE1 level of D4Z4 methylation and the FSHD score (b = −0.035, *p* = 0.140; R^2^ = 0.033).

To determine if this unexpected variability was associated with chromosome 4 or chromosome 10 alleles, we specifically measured the D4Z4 methylation status at 4q35 using the MSRE2 testing in 110 subjects with a suitable 4/2 chromosomal profile (see Methods and Appendix A). Using this 4q35-specific DNA methylation test, we still detected a highly variable D4Z4 methylation level ranging from 0% to 85% (Figure 1B). Statistical analysis revealed a very weak relationship between the level of D4Z4 methylation at 4q35 and the FSHD score (b = −0.050, p = 0.036, R^2^ = 0.057). We conclude that DNA methylation at chromosome 4q35 is highly variable in FSHD1 subjects and is not a strong indicator of disease severity.

### 2.2. Analysis of D4Z4 Methylation in FSHD1 and FSHD2

It has been reported that D4Z4 methylation level is lower in FSHD2 than in FSHD1 because all four D4Z4 arrays (2 at 4q35 and 2 at 10q26) are epigenetically dysregulated [20,21]. Originally, the threshold of 25% of total D4Z4 DNA methylation was used as a cutoff to identify FSHD2 affected subjects [22]. In a later work, the threshold was subsequently moved up to 30% [28]. In our cohort, we found that out of 34 index cases with classical FSHD phenotype carrying normal-sized D4Z4 alleles at 4q (FSHD2), 18 (52.9%) displayed CpG methylation levels higher than the 25% cutoff, assessed with the MRSE2 assay. To test whether D4Z4 methylation at the proximal *Fse*I site supports the identification of FSHD2, we measured the difference in the mean level of D4Z4 methylation obtained using MSRE1 and MSRE2 between FSHD1 (*n* = 88) and FSHD2 group (*n* = 34) of index cases (Figure 2). As presented in Table 1, our analysis did not reveal a significantly different D4Z4 methylation level (Δ MSRE1 = 4.7, *p* = 0.108; Δ MSRE2 = −3.6, p = 0.275). 

### 2.3. Analysis of D4Z4 DNA Methylation in FSHD Families

To verify whether D4Z4 methylation status can be a reliable prognostic marker in FSHD, we analyzed relatives carrying the same molecular defect at 4q35 and presenting classical FSHD (categories A1, A2, A3), or incomplete (categories B1 and B2) or complex (categories D1 and D2) phenotype or relatives with no muscle impairment (categories C1 and C2). Figure 3, Table 2, and Appendix A show the distribution of methylation percentages of 232 individuals stratified over clinical categories in two groups of subjects, probands and relatives. A similar result was obtained by using the MSRE2 assay (Figure 3B). As shown in Figure 3A and Table 2, the mean D4Z4 methylation levels detected with MSRE1 assay in subjects belonging to the different clinical categories (A, B, and D) ranges between 33% and 38.3%, including healthy subjects (category C). 

### 2.4. Evaluation of the Predictive Value of D4Z4 DNA Methylation in FSHD

To verify the value of D4Z4 DNA methylation in predicting the FSHD status, we compared D4Z4 methylation level in relatives with classical FSHD phenotype (category A) and unaffected relatives (category C). The performance of a model based on methylation at D4Z4 repeats as a discriminator between pathogenic and healthy condition was analyzed using the receiver operating characteristic (ROC) curve. The ROC curve in Figure 4 shows that D4Z4 methylation status is a poor predictor of disease status, as the calculated area under the curve (AUC) is not significantly different from 0.5 as reported in Table 3. MSRE1 analysis suggested that the D4Z4 methylation status might instead aggregate in a family pattern (data not shown) rather than being a general mark of FSHD.

### 2.5. Analysis of D4Z4 Methylation Status and Reduced Penetrance in FSHD Families

We further investigated the prognostic and diagnostic values of D4Z4 methylation in FSHD1 families with reduced penetrance (Appendix A for details). In one family, Family C, the 4q-contracted allele (4RU) was detected in eleven people. As shown in Figure 5A, subjects (II·2, III·6, III·9, IV·3, and IV·5) aged 91, 59, 57, 53, 36, and 39, respectively, did not show any sign or symptom of disease (FSHD score 0). Six (III·1, III·3, III·5, III·7, IV·1, and IV·2) presented the clinical features characteristic of FSHD (category A, FSHD score ≥1). We first performed 4q haplotyping followed by a BSS assay, which specifically analyzes the most telomeric 4qA- or 4qA-L-associated D4Z4 RU (BSSA or BSSL, respectively) in all family members carrying DRA (Figure 5 and Appendix A). The haplotyping revealed the presence of one 4qA161 allele and one 4qB163 allele in all subjects except II-1 (4qB163/4qB163), II-2 (4qA161/4qA161), and IV-3 (4qA161/4qA-L161), with the 4qA161 being 4 RUs and the other 4q chromosome of normal size (17 RU or 22 RU D4Z4 in all subjects except in III·1 (4qA161 30 RU)). To avoid diluting the values by averaging with the methylation levels of the non-contracted array in 4qA161/4qA161 subjects, we used the first quartile (Q1) for the BSS methylation status. The obtained BSS percent methylation for the Q1 of all analyzed alleles are provided in Appendix A. As shown in Figure 5, we found that among the six clinically affected subjects the Q1 D4Z4 methylation ranged between 6.3% and 28.6% and among the healthy relatives with DRAs the Q1 D4Z4 methylation ranged between 3.6% and 17.0%. Low methylation levels were detected in four affected (III·1, III·3, III·5, and IV·2) but also in four unaffected family members (III·6, III·9. IV·3, IV·5). The highest level of D4Z4 methylation in the family was observed in the affected IV·1 subject (Q1 methylation = 28.6), and the lowest (Q1 methylation = 3.6) in the unaffected III·9 subject. Thus, while this BSSA assay accurately identified all subjects in the family with a DRA as having low DNA methylation levels [27], the Q1 percent methylation did not correlate with disease presentation or severity, similar with what has been previously reported for families with reduced disease penetrance [29]. 

In a second family, Family A, two (I·1 and II·1) out of five carriers of 4qA contracted alleles (6 RU D4Z4) presented the clinical features characteristic for FSHD, with FSHD scores 7/15 and 6/15, respectively (Figure 6). Three additional carriers (II·2, II·3, and II·5) did not show any sign or symptom of disease (FSHD score = 0/15). The BSS approach again enabled us to distinguish the 4qA161 contracted allele, which has been transmitted from the affected father I·1 to his children, II·1, II·2, II·3, and II·5, from the non-contracted 4qA-L161 allele with 32 D4Z4 RU (Figure 6). Using BSS, we observed that DNA methylation of the contracted 4qA-D4Z4 alleles ranged from 8.0% to 31.3%. Again, the lowest level of methylation (8.0%) was detected in a clinically unaffected relative, II·3. Our analysis also showed that the D4Z4 methylation of the 32 RU D4Z4 allele is above 73% regardless of clinical diagnosis (Figure 6).

### 2.6. D4Z4 DNA Methylation Analysis in A Fully Penetrant FSHD1 Family

To further investigate the possibility that the D4Z4 methylation status might depend on the family background, we identified a family with severe facial, shoulder girdle, and pectoral muscle weakness and generalized atrophy (family B). The affected father had four affected children, two daughters and two sons (II·1, II·2, II·3, II·4) (Figure 7). All were wheelchair-bound from 30, 30, 38, and 18 years of age, respectively. All subjects reported age at onset in the second decade of life. Routine molecular testing and BSS analysis revealed the presence of two permissive D4Z4 alleles, one 4qA161 in the healthy range (32 RU D4Z4) and one 4qA-L161 D4Z4 with 2 RU. As shown in Figure 7, the DNA methylation of the distal 4qA D4Z4 region assessed by BSS methodology revealed high CpG DNA methylation content (≥50%) in all family members on the non-contracted (32 U) allele. However, the contracted (2 RU) 4qA-L showed a wide range of methylation levels (10%–30%) despite similar FSHD scores and the levels, again, the methylation status did not correlate with disease severity. Thus, in Family B, we failed to detect reduced D4Z4 methylation using this BSS technique in the affected family members. We concluded that in this family the D4Z4 methylation status is not associated with a very severe clinical status.

## 3. Discussion

D4Z4 hypomethylation has been associated with disease in individuals with a short D4Z4 array [20,24,26,28,29]. In recent years, D4Z4 hypomethylation has been proposed as a reliable marker and additional diagnostic tool for identifying DRA associated with FSHD and to distinguish classical FSHD from FSHD2 [22,24,27,28,29]. Several diagnostic schemes using BSS for assessing D4Z4 methylation status have been proposed [24,27]. To test the general value of methylation in a diagnostic algorithm for FSHD, we selected 122 unrelated FSHD index cases presenting facial and scapular girdle muscle weakness typical of FSHD, classified as clinical category A based on the Comprehensive Clinical Evaluation Form (CCEF) [19]. Using MSRE analysis, we observed a wide variability in the level of methylation among index cases with no association with clinical manifestation or severity of the disease. This variability was detected also when we restricted our analysis exclusively to the 4q35 alleles. Higher FSHD scores do not appear to be associated with very low levels of methylation, even in families with extreme clinical manifestation of disease, such as family B (Figure 7). Overall, our MSRE analysis failed to detect a particular D4Z4 methylation pattern in FSHD index cases. This indicates that D4Z4 methylation is not broadly predictive of disease severity, consistent with previously published studies that reported DNA hypomethylation on 4q35 DRA in FSHD family members that showed no clinical manifestation of the disease [28,29,30]. However, D4Z4 methylation status has been proposed as a valuable tool in families with reduced penetrance for identifying DRAs in apparently healthy individuals since the DRA typically has an intermediate level of D4Z4 DNA methylation in asymptomatic relatives that is significantly lower than that found in healthy relatives lacking a DRA [29]. Our BSS data of Families A, B, and C is consistent with this model.

It is also known that all FSHD affected family members do not necessarily display the typical FSHD phenotype (category A) and may belong to different clinical subcategories (from B to D) [13,18]. In addition, the presence of a high number of unaffected relatives carrying DRAs increases the difficulty in predicting a possible clinical outcome and complicates FSHD genetic counseling [16]. Therefore, we extended our methylation analysis to all available family members carrying a D4Z4 reduced allele. We analyzed 110 relatives belonging to different clinical subcategories (A to D) to test whether D4Z4 methylation might be associated with clear manifestations of disease. In this group, we had subjects with classical FSHD (categories A1, A2, A3), as well as subjects within complete (categories B1 and B2) or complex (categories D1 and D2) phenotypes or relatives with no muscle impairment (categories C1 and C2). Remarkably, the mean D4Z4 methylation levels detected in subjects belonging to the different clinical categories (A, B, and D) and healthy subjects (category C) did not permit to discriminate between the different clinical categories. Indeed, by extending the methylation analysis to affected (category A) and non-affected (category C) family members we established that the D4Z4 methylation test has a low positive predictive value for the identification of clinical FSHD.

MSRE D4Z4 methylation test was also proposed to be efficacious in distinguishing cases carrying D4Z4 reduced allele at 4q35 (FSHD1) and those carrying 4q35 alleles of normal size (FSHD2). Data from the literature suggest that total D4Z4 hypomethylation is more pronounced (≤25% across all 4 D4Z4 arrays) in FSHD2 individuals with mutations in *SMCHD1* or *DNMT3B* [22,31] compared to FSHD1 and that this epigenetic modification characterizes the FSHD2 group of patients [12,21,25]. Here, using MSRE analysis, we failed to observe a significantly different mean level of total D4Z4 DNA methylation between FSHD1 and FSHD2 subjects. Importantly, we found a group of individuals with classical FSHD phenotype carrying normal-sized 4q35 D4Z4 alleles that display CpG methylation >25%.

Therefore, given the apparent lack of predictive ability of the *Fse*I methylation data, it appears that D4Z4 methylation testing in FSHD diagnostic practice should not be recommended as predictive of disease, but it might be used for a better characterization of individual families.

Besides, both BSS analysis and MRSE data were concordant and both showed a lack of association between D4Z4 methylation level and the clinical status. As an example, in family B with very severe classical phenotype we found a D4Z4 methylation of 30%, ruling out any putative correlation between disease severity and hypomethylation. In agreement with this, we found a low level of methylation in healthy subjects that contained a DRA (i.e., subject II.3 in family A, Figure 5).

In contrast with early epigenetic studies on FSHD [32,33,34], our findings indicate that D4Z4 hypomethylation is not strictly associated with the presence of the FSHD typical clinical presentation and suggest that results of D4Z4 methylation analysis must be cautiously interpreted in respect to disease prognosis, in agreement with recent epigenetic studies [28,29]. Indeed, our study shows that subject selection based on strict clinical criteria is fundamental to define a diagnostic strategy in FSHD. In fact, in agreement with our findings, the first step of a proposed DNA methylation diagnosis of FSHD is to start with a clinical diagnosis consistent with FSHD or to be a first-degree relative of an individual diagnosed with FSHD [27]. It is possible that the inclusion of subjects without a precise clinical description might introduce bias, reducing the possibility of identifying true biomarkers.

The study also has some limitations. It would be advisable to verify these data in additional studies on cohorts of subjects in whom the standardized clinical assessment would be applied, extending the BSS analysis to a larger number of selected subjects. A general issue in epigenetic analyses is that cell models might not share the epigenetic signature of primary cells targeted by the disease. Obtaining a consistent number of muscle biopsies from index cases and relatives represent a challenge which would provide more exhaustive response to questions that are posed for clinical purposes.

## 4. Materials and Methods 

### 4.1. Participants

Our study was performed on FSHD families recruited through the Italian Clinical Network for FSHD (ICNF) (www.fshd.it). All clinical and molecular data were collected in the Miogen database and the Italian National Registry for FSHD (INRF) (www.fshd.it). Signed informed consent was obtained from each subject before inclusion in the study. 

Clinical status was ascertained using the Comprehensive Clinical Evaluation Form (CCEF), which evaluates the distribution and degree of motor impairment [19]. The CCEF has been developed by the ICNF to classify subjects based on their clinical features. The CCEF classifies: 1) subjects presenting facial and scapular girdle muscle weakness typical of FSHD (category A, subcategories A1-A3), 2) subjects with muscle weakness limited to scapular girdle or facial muscles (category B subcategories B1, B2), 3) asymptomatic/healthy subjects (category C, subcategories C1, C2), 4) subjects with myopathic phenotype presenting clinical features not consistent with FSHD canonical phenotype (D, subcategories D1, D2). The degree of clinical severity was assessed by using the FSHD clinical scale, which evaluates distinct muscle groups independently and translates disability into a number, the FSHD score. The FSHD score ranges from zero, when no objective evidence of muscle functional impairment is present, to 15, when all the muscle groups tested are severely impaired [35].

We enrolled 122 index cases presenting facial and scapular girdle muscle weakness typical of FSHD (category A, subcategories A1–A3) (Appendix A). Clinical and molecular assessments were extended to 110 family members belonging to 60 unrelated families (Figure 8 and Appendix A). In the remaining 62 index cases, the family study was not possible because they were either *de novo* DRA carriers (17 subjects) or isolated cases (45 subjects) for whom no additional relatives were available. 

Out of 122 index cases (mean age at examination of 48·5 years), 73 (59.8 %) were male. Among 110 relatives (mean age at examination of 45.3 years), 49 (44.5%) were male (Table 4).

All individuals were molecularly characterized by the routine diagnostic molecular test for the assessment of the size of D4Z4 arrays at 4q35. Through this approach, we established whether the individual presented standard allele constitution of two 4-type D4Z4 arrays on chromosome 4 and two 10-type D4Z4 arrays on chromosome 10, here termed chromosomal profile 4/2, or alleles deriving from different 4q-10q translocations and resulting in chromosomal profiles with apparently altered ratios termed 4/1 and 4/3, as shown in Appendix A.

### 4.2. Clinical Investigation

In this study, all subjects recruited were examined by applying the Comprehensive Clinical Evaluation Form (CCEF), a recently published standardized clinical tool with inter-rater reliability [19]. The CCEF consists of four sections. The first section (Evaluation Form) investigates the subject’s clinical history and the patient’s disability and assesses muscle segmental involvement. The second section includes the FSHD Evaluation Scale to calculate the FSHD clinical score (range from 0 to 15) [35]. The combination of the clinical features suggestive or not of FSHD, summarized in the Clinical Diagnostic Form, assigns patients to different Phenotypic Categories. In particular, (1) subjects presenting facial and scapular girdle muscle weakness typical of FSHD are classified as category A, subcategories A1–A3; (2) subjects with muscle weakness limited to scapular girdle or facial muscles are category B, subcategories B1, B2, respectively; (3) asymptomatic/healthy subjects are category C, subcategories C1, C2; (4) subjects with myopathic phenotype presenting clinical features not consistent with FSHD canonical phenotype are category D, subcategories D1, D2.

### 4.3. Molecular Analysis for 4q35 and 10q26

DNA was prepared from isolated lymphocytes according to standard procedures. In brief, restriction endonuclease digestion of DNA was performed in agarose plugs with the appropriate restriction enzyme: *EcoR*I, *EcoR*I/*Bln*I. Digested DNA was separated by PFGE in 1% agarose gels as previously described [10].

Allele sizes were estimated by Southern hybridization with the p13E-11 probe of 7μg of *EcoR*I-*EcoR*I/*Bln*I-digested genomic DNA extracted from peripheral blood lymphocytes, electrophoresed in 0.4% agarose gel, for 45–48 h at 35 V, alongside an 8–48 kb marker (Bio-Rad, Bio-Rad Laboratories S.r.l., CA, USA) (Appendix A). To distinguish whether the D4Z4-reduced allele originates from chromosome 10q or 4q, DNA from each index case was analyzed by *Not*I digestion and hybridized with the B31 probe [10]. Restriction fragments were detected by autoradiography or using a Typhoon Trio system (GE Healthcare Chicago, IL, US).

### 4.4. Methylation-Sensitive Restriction Enzyme (MSRE) Analysis 

DNA was prepared from isolated lymphocytes according to standard procedures. By Methylation-Sensitive Restriction Enzyme (MSRE) approach, we investigated the status of DNA methylation at CpGs positioned at the most proximal D4Z4 repeat in (1) both chromosomes 4q and chromosome 10q (MSRE1) or (2) exclusively in chromosomes 4q excluding chromosomes 10q (MSRE2) (Appendix A).

In brief for MSRE1, 5 µg of genomic DNA were digested with restriction enzyme *Bgl*II (NEB) and subsequently with the *Fse*I methylation-sensitive restriction enzyme (NEB). DNA digested with *Bgl*II/*Fse*I was separated in 0.8% agarose gel for 24 h. MSRE1 test allowed obtaining the level of D4Z4 DNA methylation in all 122 FSHD index cases and their 110 relatives. For MSRE2, 5 µg of genomic DNA was digested with restriction enzymes *Bgl*II (NEB) and with *BlnI*I (NEB) followed by *Fse*I (NEB) digestion. DNA digested with *Bgl*II/*BlnI*/*Fse*I was separated in 0.8% agarose gel for 24 h. MSRE2 was performed in the subjects presenting 4/2 chromosomal profile; 110 FSHD index cases and 102 relatives were suitable for this analysis (Table 4). Southern blotting and radioactive hybridization of MSRE1- and MSRE2-digested DNA were performed using the probe p13E-11. MSRE1- or MSRE2-digested DNA fragments were detected by autoradiography with the Typhoon Trio system (GE Healthcare), and the signal intensities were acquired and quantified with the Image Quant software. The final result is presented in percentage of methylation calculated as the ratio between digested (hypomethylated) and undigested (methylated) DNA. 

### 4.5. Sodium Bisulfite Sequencing (BSS)-DNA Methylation Analysis

Genomic DNA samples extracted from lymphocytes were bisulfite converted, amplified using primers specific for the distal 4qA and 4qA-Long (4qA-L) D4Z4 repeat regions, and then sequenced, as described [27]. This BSS assay is specific for analyzing the DNA methylation status of the most distal D4Z4 unit on permissive 4qA chromosomes by utilizing polymorphisms in the primers that are exclusive to 4qA and not found in 10qA or 4qB. The 4qA BSS assay analyzes 56 CpGs in the most telomeric D4Z4 RU on 4qA-containing chromosomes. A fraction of chromosomes characterized as 4qA are an allelic variant termed 4qA-L; these contain an additional 2 kb of D4Z4 sequence at the distal repeat although the A-type subtelomere is unchanged. Thus, the 4qA-L BSS assay utilizes the same 4qA-specific reverse BS-PCR primers as the 4qA assay but analyzes a distinct set of 30 CpGs in the distal repeat on 4qA-L chromosomes, as described in Reference [27]. To account for differences in the number of assayable 4q alleles in subjects (1 or 2), DNA methylation from this BSS assay is reported in methylation percentage quartiles for each linearly sequenced chromosome, calculated such that with n chromosomes ordered highest methylation to lowest methylation, the first quartile (Q1) is methylation of the n/4-th chromosome. If n/4 is not an integer, then interpolation is used between the nearest values. Thus, the greater the n chromosomes assayed, the higher the confidence in the presented Q1 values. Q1 value less than 25% for 4qA assay and Q1 value less than 30% for 4qA-L assay are characterized as FSHD.

### 4.6. Statistical Analysis

Characteristics of subjects were compiled using established descriptive statistics. To quantify the possible relationship between continuous variables (D4Z4 methylation level, FSHD score), we carried out linear regression analysis. To compare means (D4Z4 methylation percentage, FSHD score), we used *t*-test and analysis of variance (ANOVA, graphPad Prism version 8.00 for Windows, GraphPad Software, San Diego, CA, USA). To establish the relationship between disease status and methylation percentage, we constructed receiver operating characteristic (ROC graphPad Prism version 8.00 for Windows, GraphPad Software, San Diego, CA, USA) curves.

### 4.7. Ethical Statement

The study was approved by the Local Ethics Committees of all participating Institutions. The study was performed in accordance with the ethical standards laid down in the 1964 Declaration of Helsinki and its later amendments.

## 5. Conclusions

In conclusion, the variability in D4Z4 methylation observed in our purposefully selected cohort of clinically homogeneous patients and their healthy relatives, as well as the detection of pronounced D4Z4 hypomethylation below the 25% threshold in only half of phenotypic FSHD2 cases, show that this epigenetic mark has a low predictive clinical value. Therefore, D4Z4 methylation can accurately identify DRAs in FSHD families. DNA methylation analysis alone is insufficient as a prognostic and diagnostic tool for disease presentation and severity and should be properly complemented with a standardized clinical assessment and proper family studies.

## Figures and Tables

**Figure 1 ijms-21-02635-f001:**
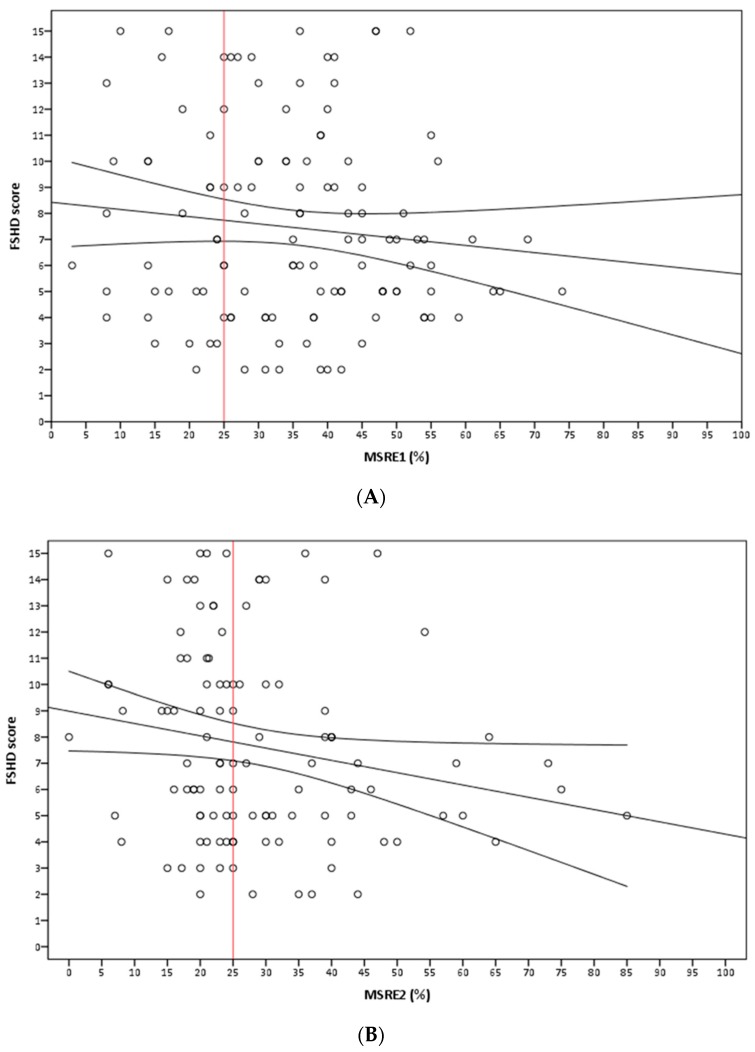
Assessment of DNA methylation level in facioscapulohumeral muscular dystrophy (FSHD) index cases using MRSE1 (A) and MRSE2 (B) approaches: (**A**) The crude relationship between the two parameters is not significant (b = −0.028, *p* = 0.236; R^2^ = 0.012). The strength of the relationship does not improve when adjusting for the age of patient, as we find no significant association between level of D4Z4 methylation and FSHD score (b = −0.035, *p* = 0.140; R^2^ = 0.033). (**B**) The crude relationship is very weak but statistically significant (b = −0.047, *p* = 0.048; R^2^ = 0.036). The strength of the relationship improves adjusting for the age of patient (b=−0.050, *p* = 0.036, R^2^ = 0.057).

**Figure 2 ijms-21-02635-f002:**
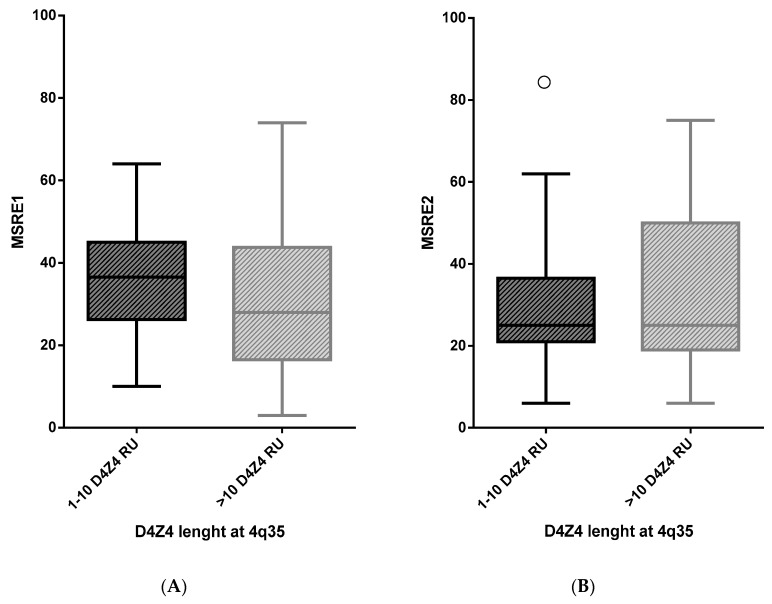
Comparison of D4Z4 DNA methylation level between FSHD index cases carrying 1–10 D4Z4 repeat units (RU) (FSHD1) or more than 10 D4Z4 RU (FSHD2): A box plot of D4Z4 methylation level assessed with MSRE1 (**A**) and MSRE2 (**B**) approaches in carriers of 1–10 or >10 D4Z4 units. The circle indicates an outlier value.

**Figure 3 ijms-21-02635-f003:**
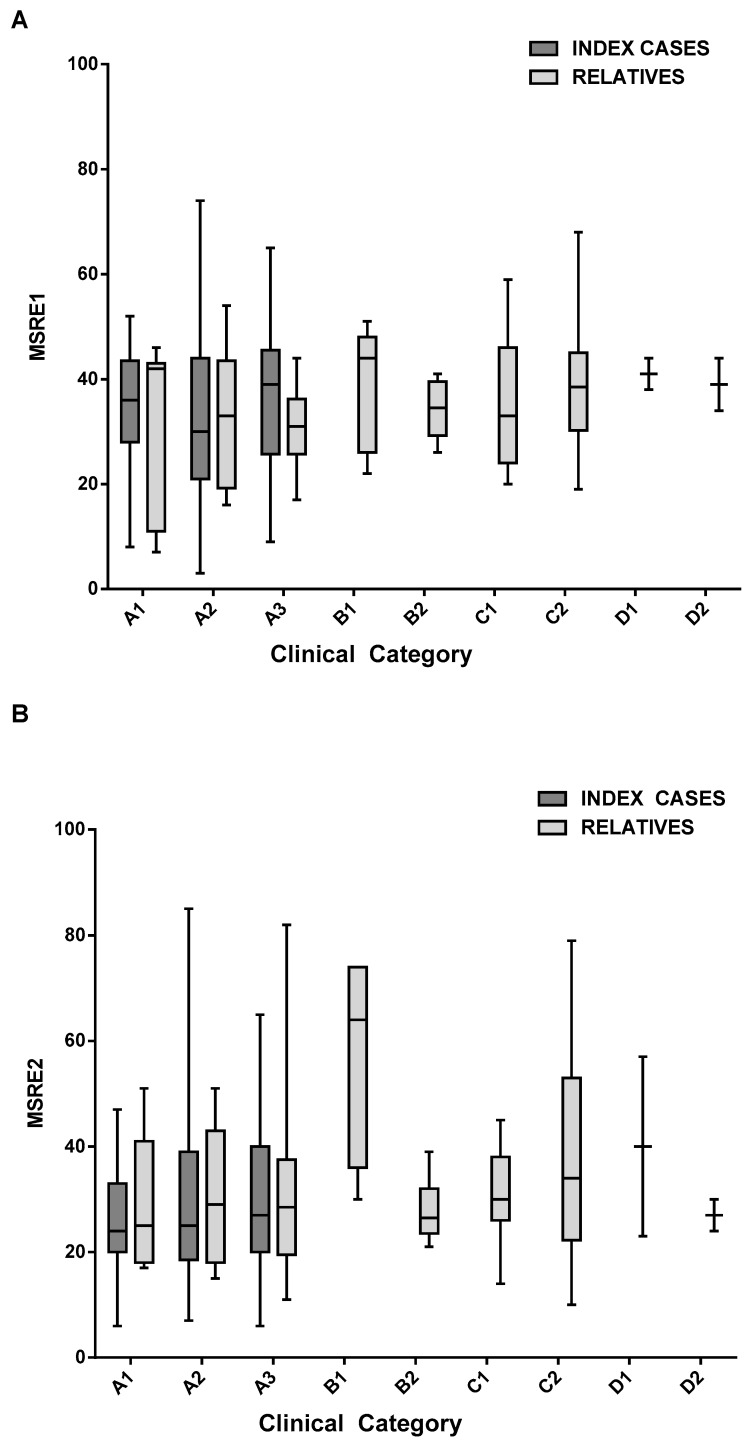
Evaluation of D4Z4 methylation levels in subjects belonging to different clinical categories: A box plot of D4Z4 methylation levels assessed with MSRE1 assay (**A**) and MSRE2 (**B**) in probands and relatives stratified over clinical categories (A, B, C, and D).

**Figure 4 ijms-21-02635-f004:**
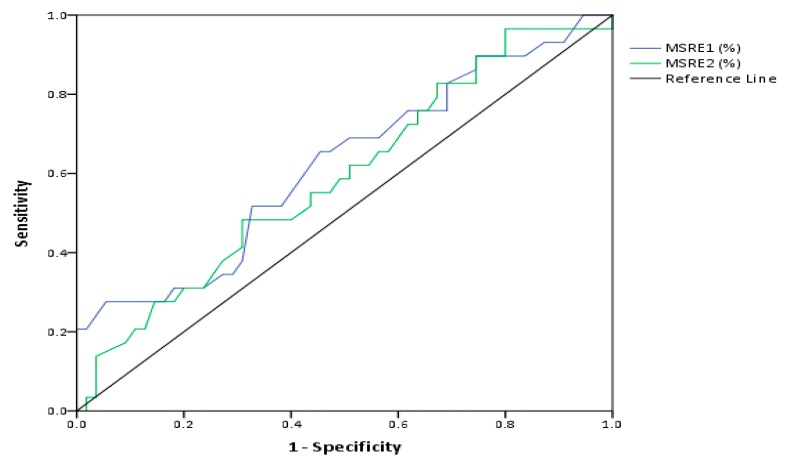
Evaluation of the efficacy of D4Z4 methylation status as a marker for classifying diseased and non-diseased individuals. The receiver operating characteristic (ROC) curve was plotted to analyze the values of methylation at D4Z4 as a discriminator between affected and unaffected individuals.

**Figure 5 ijms-21-02635-f005:**
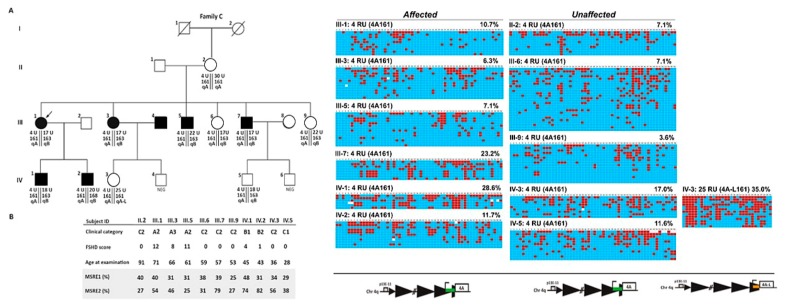
Analysis of the D4Z4 methylation status in Family C. (**A**) Pedigree of family C. The genetic profile (**A**) or the clinical and epigenetic features (**B**) of each individual are reported. Filled symbols represent affected people. (**C**) Genomic DNAs from the selected individuals were analyzed using the 4qA or 4qA-L sodium bisulfite sequencing (BSS) assay. The percent DNA methylation for the Q1 is indicated. Red boxes indicate methylated CpGs, blue boxes indicate unmethylated CpGs, and white boxes indicate no CpG at the expected site.

**Figure 6 ijms-21-02635-f006:**
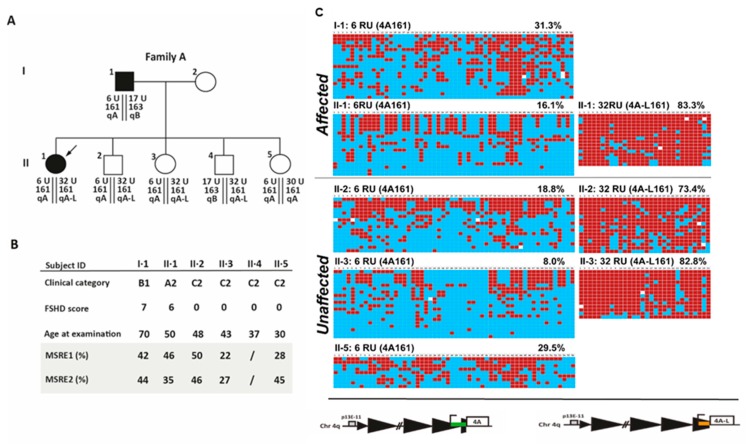
Analysis of the D4Z4 methylation status in family A. (**A**) Pedigree of family A. The genetic profile (**A**) or the clinical and epigenetic features (**B**) of each individual are reported. Filled symbols represent affected people (**C**) Genomic DNAs from the selected individuals were analyzed using the 4qA or 4qA-L BSS assay. The percentage DNA methylation for the Q1 is indicated. Red boxes indicate methylated CpGs, blue boxes indicate unmethylated CpGs, and white boxes indicate no CpG at the expected site.

**Figure 7 ijms-21-02635-f007:**
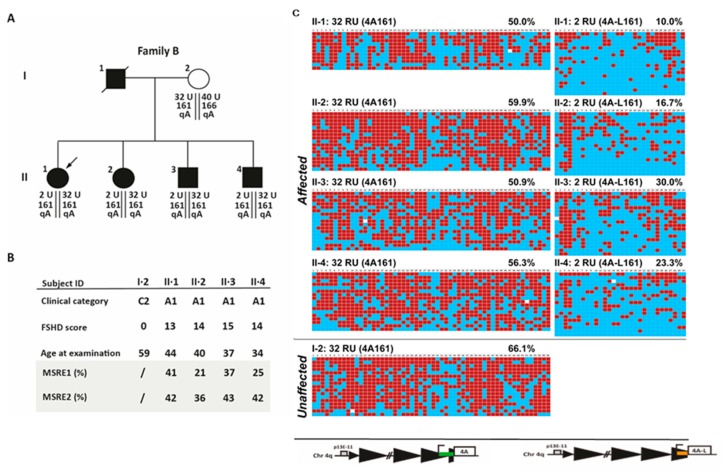
Analysis of the D4Z4 methylation status B. (**A**) Pedigree of family B. The genetic profile (**A**) or the clinical and epigenetic features (**B**) of each individual are reported. Filled symbols represent affected people (**C**) Genomic DNAs from the selected individuals were analyzed using the 4qA BSS assay. The percentage of DNA methylation for the Q1 is indicated. Red boxes indicate methylated CpGs, blue boxes indicate unmethylated CpGs, and white boxes indicate no CpG at the expected site.

**Figure 8 ijms-21-02635-f008:**
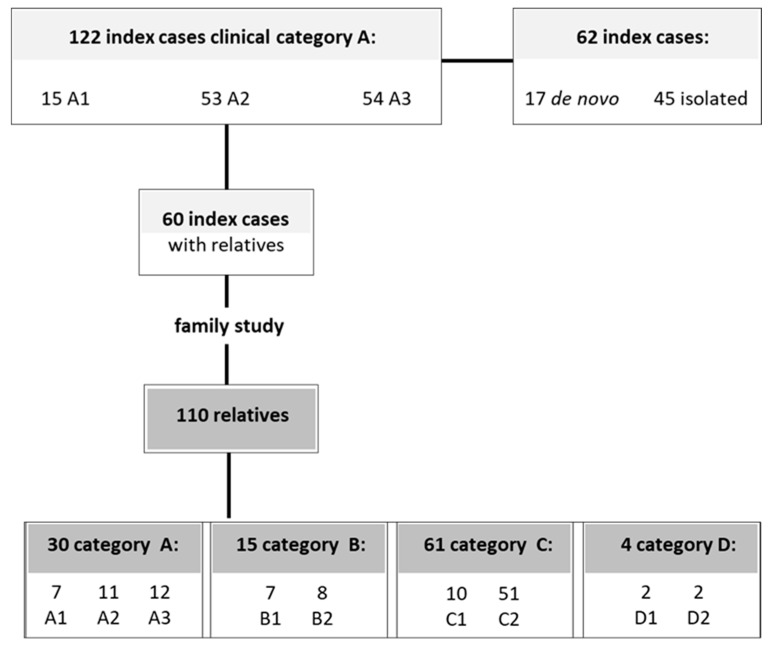
Patients recruitment and their clinical status.

**Table 1 ijms-21-02635-t001:** Mean D4Z4 methylation level and difference (Δ) in mean D4Z4 methylation between two groups reported in Figure 2.

FSHD Index Cases	D4Z4 (RU)	Number of Subjects	Mean D4Z4 Methylation Level (%)	SD	SE	Difference in Mean D4Z4	*p*-Value
						Methylation between Two Groups	
						Δ	
MSRE1	1–10	88	36.3	11.9	1.3	4.7	0.108
	>10	34	31.6	19.5	3.3		
MSRE2	1–10	82	28.0	12.0	1.3	−3.6	0.275
	>10	28	31.6	21.5	4.1		

SE = Standard error; SDS = Standard deviation.

**Table 2 ijms-21-02635-t002:** The mean D4Z4 methylation values for the MRSE1 assay graphed in Figure 3A and the 95% confidence interval (CI).

MSRE1	Clinical Category	Mean D4Z4 Methylation Level	95% CI
index cases	A1	34.3	[28, 40.6]
	A2	33.2	[33.2, 37.6]
	A3	37	[33.4, 40.8]
relatives	A1	33	[16.2, 49.8]
	A2	33.2	[23.5, 42.9]
	A3	31.1	[26.4, 35.9]
	B1	37.3	[26.1, 48.5]
	B2	34.1	[29.5, 38.7]
	C1	37.1	[27.7, 46.5]
	C2	38.3	[35.2, 41.4]
	D1	41	[35.2, 46.8]
	D2	39	[29.2, 48.8]

**Table 3 ijms-21-02635-t003:** The area under the curve (AUC) threshold of the model.

Test Result Variables	AUC	SE	Asymptotic 95% CI
MSRE1	0.622	0.1	[0.5,0.8]
MSRE2	0.592	0.1	[0.5,0.7]

AUC = Area under curve; SE = Standard Error; CI = 95% Confidence Interval.

**Table 4 ijms-21-02635-t004:** Baseline characteristics of selected participants.

	Number of Subjects	Sex		Age at Examination		D4Z4 (RU)	
		male	female	mean	SD	1–10	>10
index cases	122	73 (59.8%)	49 (40.1%)	48.5	18.5	88	34
relatives	110	49 (44.5%)	61 (55.4%)	45.3	15.5	107	2

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
