# Peer review of "Interpretation of the Epigenetic Signature of Facioscapulohumeral Muscular Dystrophy in Light of Genotype-Phenotype Studies"

_ijms, 2020, doi:10.3390/ijms21072635_

Round 1

Reviewer 1 Report

This study by Nikolic et al. investigates the correlation between methylation status at the 4q and 10q D4Z4 arrays with disease severity, type, and penetrance in patients with facioscapulohumeral muscular dystrophy (FSHD). The authors also evaluate the utility of methylation status in the prediction of FSHD status. A cohort of index patients were used, roughly half of which had data from family members included in the study. The work is based on an interesting question in the field (that of genotype-phenotype correlation), and was thorough to some extent in the use of analytical tools to extract information from the clinical and molecular data obtained. While certainly important work, some concerns have to be addressed. These will also help improve the readability of the manuscript, as well as understanding to non-specialist readers:

  1. Indicate what the differences between subcategories are, e.g. how different is A1 from A2, etc.
  2. Figure 1 and lines 137-141 require more clarification in terms of presentation, as it is hard to follow at the moment. Perhaps a better flowchart will help. Also, when the numbers on the right of Figure 1 are added, the result is 110 and not 109 as stated in line 138. Please confirm or correct if needed.
  3. Line 156: Just a thought, but is the use of DNA from lymphocytes a factor in the overall lack of genotype-phenotype correlations observed in the study? If so, please discuss this in the manuscript.
  4. Supplementary Figures 1A and 1B should be switched, since 1B is introduced earlier in the text than 1A. Figure S1B is also confusing – is this result from after or before MSRE? The figure legend for this panel seems to indicate that it is before MSRE, and hence must be revised if needed.
  5. Line 169: “most centromeric D4Z4 repeat”, for consistency, should the terms proximal/distal be used here?
  6. Line 175: “AvrII” is not apparent in Supplementary Figure 1A.
  7. Section 3.1: Were any differences observed for this analysis when patients were stratified, e.g. in terms of sex, or age of onset, or other clinical parameters?
  8. Provide a citation for the sentence in lines 248-249.
  9. Figure 3A, right panel has some symbols that weren’t explained in the legend (asterisk, and open circle). Please explain.
  10. There are some tables that are incorporated in figures in this paper (e.g. Figure 3B, 4B, 5B, etc.). It would be better to indicate that a table is a table and not a figure.
  11. Is there a corresponding MSRE2 result for Section 3.3?
  12. Supplementary Tables have to be captioned, and better referenced in the text.
  13. Would a correlation be better expected if the expression of downstream targets of DUX4 were evaluated instead? This may work as a functional read-out of methylation status at the 4qA D4Z4 array. The limitations of this study also have to be better explained in the Discussion.
  14. Proofreading is required:
    1. Line 112: “10” to “10q”
    2. Line 170: change “chromosomes” to singular form
    3. Line 241: “the” should be capitalized, line 242: add “when” before “adjusting”
    4. Line 355: “lower” to “lowest”

Author Response

Comments and Suggestions for Authors

  1. Indicate what the differences between subcategories are, e.g. how different is A1 from A2, etc.

R: We thank Reviewer 1 for the observation. The phenotypic and clinical features observed in FSHD patients and the usage of the Comprehensive Clinical Evaluation Form (CCEF) were described in the work by Ricci et al., 2016 in which the CCEF was presented and tested for inter-rater reliability. To make the consultation easier, we introduced  the clinical evaluation methodology in the Methods section.

  1. Figure 1 and lines 137-141 require more clarification in terms of presentation, as it is hard to follow at the moment. Perhaps a better flowchart will help. Also, when the numbers on the right of Figure 1 are added, the result is 110 and not 109 as stated in line 138. Please confirm or correct if needed.

R:  We thank Reviewer 1 for her/her suggestion of making Figure 1 clearer. We modified it accordingly.

The number was corrected in the text.

  1. Line 156: Just a thought, but is the use of DNA from lymphocytes a factor in the overall lack of genotype-phenotype correlations observed in the study? If so, please discuss this in the manuscript.

R: We agree with Reviewer 1 that lymphocytes might not share the epigenetic signature of muscle cells, the primary target of FSHD. However obtaining such a large number of muscle biopsies from index cases and relatives requests a great effort in terms of time and patients engagement. DNA from lymphocytes is routinely used for diagnosis in genetic diseases, despite it might not represent the perfect match with the genomic sequence of the target tissue because of its availability with not invasive procedure. In FSHD it has been reported that the D4Z4 DNA methylation in lymphocytes matches the methylation status in myoblasts [1].

  1. Supplementary Figures 1A and 1B should be switched, since 1B is introduced earlier in the text than 1A. Figure S1B is also confusing – is this result from after or before MSRE? The figure legend for this panel seems to indicate that it is before MSRE, and hence must be revised if needed.

R: We thank Reviewer 1 for the observation. Supplementary Figure 1B (now 1A) shows the different chromosomal profiles observed  after PFGE electrophoresis of DNA samples digested with the restriction enzyme EcoRI (E) or EcoRI / BlnI (E / B), with the aim to illustrate the DNA profiles used for MRSEI or MRSEII analysis.  As reported in the Materials and Methods section, DNA was digested with BglII/FseI for MRSEI and with BglII /BlnII followed by FseI for MRSEII assays. For clarity, the Legend of Figure 1 was changed accordingly.

  1. Line 169: “most centromeric D4Z4 repeat”, for consistency, should the terms proximal/distal be used here?

R: Thank you, the word was corrected.

  1. Line 175: “AvrII” is not apparent in Supplementary Figure 1A.

R: We thank Reviewer 1 for the observation . We apologise for the error. AvrII is the alternative name  for the BlnI enzyme. The term AvrII was replaced with BlnI in text and Figures.

  1. Section 3.1: Were any differences observed for this analysis when patients were stratified, e.g. in terms of sex, or age of onset, or other clinical parameters?

R: We compared the methylation status on the basis of gender (males versus females) and severity of disease expression expressed as FSHD score (FSHD score 0 no muscle impairment, 1-2 mild muscle impairment, 3-7 moderate muscle impairment, 8-15 muscle impairment). We observed no significative differences in the methylation levels.

  1. Provide a citation for the sentence in lines 248-249.

R: We thank Reviewer 1 for the suggestion. Citations were inserted in line 255, Refs :

[20] van Overveld, P.G.M.; Lemmers, R.J.F.L.; Sandkuijl, L.A.; Enthoven, L.; Winokur, S.T.; Bakels, F.; Padberg, G.W.; van Ommen, G.-J.B.; Frants, R.R.; van der Maarel, S.M. Hypomethylation of D4Z4 in 4q-linked and non-4q-linked facioscapulohumeral muscular dystrophy. Nat. Genet. 2003, 35, 315–317.

[21] de Greef, J.C.; Lemmers, R.J.L.F.; van Engelen, B.G.M.; Sacconi, S.; Venance, S.L.; Frants, R.R.; Tawil, R.; van der Maarel, S.M. Common epigenetic changes of D4Z4 in contraction-dependent and contraction-independent FSHD. Hum. Mutat. 2009, 30, 1449–1459.

  1. Figure 3A, right panel has some symbols that weren’t explained in the legend (asterisk, and open circle). Please explain.

R: We thank Reviewer 1 for the observation.  The asterisk was a misprint, we apologise for the mistake. The circle corresponds to an outlayer value, as we reported in Legend for Figure 3, Line 271.

  1. There are some tables that are incorporated in figures in this paper (e.g. Figure 3B, 4B, 5B, etc.). It would be better to indicate that a table is a table and not a figure.

R: We thank Reviewer 1 for the suggestion. The Figures were changed and the Tables were separated from Figures 3, 4 and 5 and inserted as Tables 2, 3 and 4 respectively. They were cited in the text accordingly in lines 262, 282, 285 and 318.

  1. Is there a corresponding MSRE2 result for Section 3.3?

R: We thank Reviewer 1 for the suggestion. We inserted, the graph representing the results for MSRE2 assay in Figure4, part B. Data were discussed in Section 3.3, line 284.

  1. Supplementary Tables have to be captioned, and better referenced in the text.

R: We thank Reviewer 1 for the suggestion. We added the Caption for Supplementary tables (lane 484), and Supplementary tables were cited in the text in lines 137, 138, 226, 282 336 and 347.

  1. Would a correlation be better expected if the expression of downstream targets of DUX4 were evaluated instead? This may work as a functional read-out of methylation status at the 4qA D4Z4 array. The limitations of this study also have to be better explained in the Discussion.

R: We thank Reviewer 1 for the observations

  1. a) The present work focuses on the evaluation of the percentage of CpG methylation and its significance in clinical practice in FSHD. We did not intent to discuss the actual model proposed to explain FSHD pathogenesis. Indeed, there are multiple epidemiological data suggesting that D4Z4 reduction associated with a permissive haplotype for the expression of DUX4 retrogene is not sufficient to explain and predict disease. For instance, studies from the Italian National Registry for FSHD gathers a large number of individuals with in-depth clinical analysis ([4–8] and Ruggiero et al., 2020, accepted for publications) that are consistent with other genotype-phenotype studies performed in other population [9–16] highlight the vast heterogeneity associated with FSHD and encourage the medical community to not oversimplify a complex pathology. We would like to highlight that our study showed that D4Z4 methylation status reflect the number of D4Z4 copies with no correlation with the clinical status.

  1. b) Sentences discussing possible limitations of the study were inserted in the Discussion, lines 458:

“The study also has some limitations. It would be advisable to verify these data in additional studies on cohorts of subjects in whom the standardized clinical assessment would be applied, extending the BSS analysis to larger number of selected subjects. A general issue in epigenetic analyses is that cell models might not share the epigenetic signature of primary cells targeted by the disease. Obtaining a consistent number of muscle biopsies from index cases and relatives represents a challenge which would provide more exhaustive response to questions that are pose for clinical purposes. “

  1. Proofreading is required:
    1. Line 112: “10” to “10q”
    2. Line 170: change “chromosomes” to singular form
    3. Line 241: “the” should be capitalized, line 242: add “when” before “adjusting”
    4. Line 355: “lower” to “lowest”

R: We apologise with Reviewer 1 for the typing errors which have been corrected

CITED LITERATURE

  1. Jones, T.I.; Himeda, C.L.; Perez, D.P.; Jones, P.L. Large family cohorts of lymphoblastoid cells provide a new cellular model for investigating facioscapulohumeral muscular dystrophy. Neuromuscul. Disord. 2017, 27, 221–238.
  2. van Overveld, P.G.M.; Lemmers, R.J.F.L.; Sandkuijl, L.A.; Enthoven, L.; Winokur, S.T.; Bakels, F.; Padberg, G.W.; van Ommen, G.-J.B.; Frants, R.R.; van der Maarel, S.M. Hypomethylation of D4Z4 in 4q-linked and non-4q-linked facioscapulohumeral muscular dystrophy. Nat. Genet. 2003, 35, 315–317.
  3. de Greef, J.C.; Lemmers, R.J.L.F.; van Engelen, B.G.M.; Sacconi, S.; Venance, S.L.; Frants, R.R.; Tawil, R.; van der Maarel, S.M. Common epigenetic changes of D4Z4 in contraction-dependent and contraction-independent FSHD. Hum. Mutat. 2009, 30, 1449–1459.
  4. Scionti, I.; Greco, F.; Ricci, G.; Govi, M.; Arashiro, P.; Vercelli, L.; Berardinelli, A.; Angelini, C.; Antonini, G.; Cao, M.; et al. Large-scale population analysis challenges the current criteria for the molecular diagnosis of fascioscapulohumeral muscular dystrophy. Am. J. Hum. Genet. 2012, 90, 628–635.
  5. Ricci, G.; Scionti, I.; Sera, F.; Govi, M.; D’Amico, R.; Frambolli, I.; Mele, F.; Filosto, M.; Vercelli, L.; Ruggiero, L.; et al. Large scale genotype–phenotype analyses indicate that novel prognostic tools are required for families with facioscapulohumeral muscular dystrophy. Brain 2013, 136, 3408–3417.
  6. Nikolic, A.; Ricci, G.; Sera, F.; Bucci, E.; Govi, M.; Mele, F.; Rossi, M.; Ruggiero, L.; Vercelli, L.; Ravaglia, S.; et al. Clinical expression of facioscapulohumeral muscular dystrophy in carriers of 1-3 D4Z4 reduced alleles: Experience of the FSHD Italian National Registry. BMJ Open 2016, 6, 1–10.
  7. Ricci, G.; Ruggiero, L.; Vercelli, L.; Sera, F.; Nikolic, A.; Govi, M.; Mele, F.; Daolio, J.; Angelini, C.; Antonini, G.; et al. A novel clinical tool to classify facioscapulohumeral muscular dystrophy phenotypes. J. Neurol. 2016, 263, 1204–1214.
  8. Scionti, I.; Fabbri, G.; Fiorillo, C.; Ricci, G.; Greco, F.; D’Amico, R.; Termanini, A.; Vercelli, L.; Tomelleri, G.; Cao, M.; et al. Facioscapulohumeral muscular dystrophy: New insights from compound heterozygotes and implication for prenatal genetic counselling. J. Med. Genet. 2012, 49, 171–178.
  9. Salort-Campana, E.; Nguyen, K.; Bernard, R.; Jouve, E.; Solé, G.; Nadaj-Pakleza, A.; Niederhauser, J.; Charles, E.; Ollagnon, E.; Bouhour, F.; et al. Low penetrance in facioscapulohumeral muscular dystrophy type 1 with large pathological D4Z4 alleles: a cross-sectional multicenter study. Orphanet J. Rare Dis. 2015, 10, 2.
  10. Sakellariou, P.; Kekou, K.; Fryssira, H.; Sofocleous, C.; Manta, P.; Panousopoulou, A.; Gounaris, K.; Kanavakis, E. Mutation spectrum and phenotypic manifestation in FSHD Greek patients. Neuromuscul. Disord. 2012, 22, 339–349.
  11. Goto, K.; Nishino, I.; Hayashi, Y.K. Very low penetrance in 85 Japanese families with facioscapulohumeral muscular dystrophy 1A. J. Med. Genet. 2004, 41, 12e – 12.
  12. Tonini, M.M.O.; Passos-Bueno, M.R.; Cerqueira, A.; Matioli, S.R.; Pavanello, R.; Zatz, M. Asymptomatic carriers and gender differences in facioscapulohumeral muscular dystrophy (FSHD). Neuromuscul. Disord. 2004, 14, 33–8.
  13. NAKAGAWA, M.; MATSUZAKI, T.; HIGUCHI, I.; FUKUNAGA, H.; INUI, T.; NAGAMITSU, S.; YAMADA, H.; ARIMURA, K.; OSAME, M. Facioscapulohumeral Muscular Dystrophy: Clinical Diversity and Genetic Abnormalities in Japanese Patients. Intern. Med. 1997, 36, 333–339.
  14. Statland, J.M.; McDermott, M.P.; Heatwole, C.; Martens, W.B.; Pandya, S.; van der Kooi, E.L.; Kissel, J.T.; Wagner, K.R.; Tawil, R. Reevaluating measures of disease progression in facioscapulohumeral muscular dystrophy. Neuromuscul. Disord. 2013, 23, 306–312.
  15. Ricci, G.; Cammish, P.; Siciliano, G.; Tupler, R.; Lochmuller, H.; Evangelista, T. Phenotype may predict the clinical course of facioscapolohumeral muscular dystrophy. Muscle and Nerve 2019, 59, 711–713.
  16. Lin, F.; Wang, Z.-Q.; Lin, M.-T.; Murong, S.-X.; Wang, N. New Insights into Genotype-phenotype Correlations in Chinese Facioscapulohumeral Muscular Dystrophy. Chin. Med. J. (Engl). 2015, 128, 1707–1713.

Reviewer 2 Report

  1. Figure 1 can be redesigned to more clearly show that the 122 index cases were subdivided into the 60 with family study possible and the 62 with whom it was not. It is confusing as it is right now. 
  2. Why is investigating DNA methylation at the CpG of the most centromeric D4Z4 repeat a valid approach? That is, are there any known differences in CpG methylation across repeats, position-wise, in the 4q D4Z4 array in FSHD patients?
  3. Page 6, line 219: Confirm if "MRSE2" and not "MRSE1" was analyzed.
  4. Have other combinations of categories been tested for the ROC curve analysis, e.g. category A + B vs category C?
  5. Discuss possible limitations of the CCEF as they relate to the findings of the study.

Author Response

Comments and Suggestions for Authors

  1. Figure 1 can be redesigned to more clearly show that the 122 index cases were subdivided into the 60 with family study possible and the 62 with whom it was not. It is confusing as it is right now.

R:  We thank Reviewer 2 for her/her suggestion of making Figure 1 clearer. We modified it accordingly.

  1. Why is investigating DNA methylation at the CpG of the most centromeric D4Z4 repeat a valid approach? That is, are there any known differences in CpG methylation across repeats, position-wise, in the 4q D4Z4 array in FSHD patients?

R: We thank Reviewer 2 for her/her observation. The FseI site most proximal to the D4Z4 array was considered as the most sensitive for the methylation assay with an estimated DNA methylation of 70-80% in healthy individuals and a significant decrease in FSHD1 patients [1,2]. The MSRE1 assay evaluates only the few CpGs detected by the various of MSREs without distinction between the D4Z4 repeat array on 4q and 10q chromosome, while the MRSE2 assay investigates the 4q specific-D4Z4 methylation status. We and others [3,4] observed a dynamic methylation pattern within the 3.3 Kb region representing each D4Z4 array. In particular, the 5’ (centromeric, containing the FseI site) and 3’ region (telomeric) of the sequence appear to be generally hypomethylated with respect to the DUX4 hypothetical promoter (mid region) which shows high levels of CpG methylation ([4]and personal observation). DNA hypomethylation is also observed in carriers of a 4q35 reduced allele in the D4Z4 most distal repeat at the 3’ end of the pLAM region [5,6].  This region has been assessed in the BSS analysis of selected families reported in this work.

  1. Page 6, line 219: Confirm if "MRSE2" and not "MRSE1" was analyzed.

R: Yes, we confirm that we analysed MRSE2.  For clarity, we changed the sentence referring to the corresponding Figure, in line 227.

“ Remarkably, through MRSE2 assay, Figure 2B, we found that…..”

  1. Have other combinations of categories been tested for the ROC curve analysis, e.g. category A + B vs category C?

R: Thank you for the suggestion. It would be a very interesting analysis but the purpose of the reported test was to clarify the value of methylation as proxy indicator of the classical FSHD disease status (category A).

  1. Discuss possible limitations of the CCEF as they relate to the findings of the study.

R: In the past 6 years, since the CCEF has been used by the Italian Clinical Network for FSHD in the routine diagnostic evaluation we have accrued more evidences of the usefulness of the CCEF ([7]  and Ruggiero et al 2010, Ricci et al under revision).

CITED LITERATURE

  1. van Overveld, P.G.M.; Lemmers, R.J.F.L.; Sandkuijl, L.A.; Enthoven, L.; Winokur, S.T.; Bakels, F.; Padberg, G.W.; van Ommen, G.-J.B.; Frants, R.R.; van der Maarel, S.M. Hypomethylation of D4Z4 in 4q-linked and non-4q-linked facioscapulohumeral muscular dystrophy. Nat. Genet. 2003, 35, 315–317.
  2. de Greef, J.C.; Lemmers, R.J.L.F.; van Engelen, B.G.M.; Sacconi, S.; Venance, S.L.; Frants, R.R.; Tawil, R.; van der Maarel, S.M. Common epigenetic changes of D4Z4 in contraction-dependent and contraction-independent FSHD. Hum. Mutat. 2009, 30, 1449–1459.
  3. Gaillard, M.-C.; Roche, S.; Dion, C.; Tasmadjian, A.; Bouget, G.; Salort-Campana, E.; Vovan, C.; Chaix, C.; Broucqsault, N.; Morere, J.; et al. Differential DNA methylation of the D4Z4 repeat in patients with FSHD and asymptomatic carriers. Neurology 2014, 83, 733–742.
  4. Huichalaf, C.; Micheloni, S.; Ferri, G.; Caccia, R.; Gabellini, D. DNA methylation analysis of the macrosatellite repeat associated with FSHD muscular dystrophy at single nucleotide level. PLoS One 2014, 9.
  5. Jones, T.I.; King, O.D.; Himeda, C.L.; Homma, S.; Chen, J.C.J.J.; Beermann, M. Lou; Yan, C.; Emerson, C.P.; Miller, J.B.; Wagner, K.R.; et al. Individual epigenetic status of the pathogenic D4Z4 macrosatellite correlates with disease in facioscapulohumeral muscular dystrophy. Clin. Epigenetics 2015, 7, 37.
  6. Salsi, V.; Magdinier, F.; Tupler, R. Does DNA Methylation Matter in FSHD? Genes (Basel). 2020, 11, 258.
  7. Ricci, G.; Cammish, P.; Siciliano, G.; Tupler, R.; Lochmuller, H.; Evangelista, T. Phenotype may predict the clinical course of facioscapolohumeral muscular dystrophy. Muscle and Nerve 2019, 59, 711–713.